# Lateral vegetation growth rates exert control on coastal foredune "hummockiness" and coalescing time

Evan B. Goldstein[1], Laura J. Moore[1], Orencio Durán Vinent[2]

[1] Department of Geological Sciences, University of North Carolina at Chapel Hill, 104 South Rd, Mitchell Hall, Chapel Hill, NC 27599 USA

[2] Department of Physical Sciences, Virginia Institute of Marine Science, College of William and Mary, PO Box 1346, Gloucester Point, Virginia 23062

*Correspondence to*: Evan B. Goldstein (evan.goldstein@unc.edu)

**Abstract.** Coastal foredunes form along sandy, low-sloped coastlines and range in shape from continuous dune ridges to hummocky features, which are characterized by alongshore-variable dune crest elevations. Initially scattered dune-building plants and species that grow slowly in the lateral direction have been implicated as a cause of foredune hummockiness. Our goal in this work is to explore how the initial configuration of vegetation and vegetation growth characteristics control the development of hummocky coastal dunes including the 'maximum hummockiness' of a given dune field. We find that given sufficient time and absent external forcing, hummocky foredunes coalesce to form continuous dune ridges. Model results yield a predictive rule for the timescale of coalescing and the height of the coalesced dune that depends on initial plant dispersal and two parameters that control the lateral and vertical growth of vegetation, respectively. Our findings agree with previous observational and conceptual work — whether or not hummockiness will be maintained depends on the time scale of coalescing relative to the recurrence interval of high water events that reset dune-building in low areas between hummocks. Additionally, our model reproduces the observed tendency for foredunes to be hummocky along the southeast coast of the U.S. where lateral vegetation growth rates, and thus coalescing times, are likely longer.

## 1 Introduction

Vegetated coastal foredunes display various morphologies in the alongshore direction, ranging on a spectrum from continuous to hummocky (i.e., varying in dune crest elevation). Examples of hummocky foredunes from Fort Fisher State Recreation Area, NC, U.S. are shown in Figure 1. As described below, three explanations have been used (separately and in conjunction) to explain the existence of hummocky vegetated foredunes at a given site — initial configuration (i.e., spatial distribution) of plants, the rate of plant lateral expansion, and forcing or boundary conditions that control the pace and style of the biophysical feedback that gives rise to coastal dune growth.

New coastal dunes can be initiated when there is sufficient cross-shore width seaward of the existing foredune for plants to colonize (e.g., Hesp, 2002), or when elevated water levels destroy existing dunes. The presence of plants causes the deposition of sand (e.g., Hesp, 1989; Arens 1996; Kuriyama et al. 2005), leading to the formation of small dunes (Hesp, 1981; Pye, 1983). These incipient dunes have a typology that depends on the mechanism (plant, seed, rhizome, flotsam, etc.)

and alongshore continuity of plant establishment (Hesp, 1989; Hesp 2002; Hesp and Walker 2013), and variability in the location where plants initially grow can cause the formation of hummocky dunes. For example, Godfrey (1977) noted that in some settings vegetation initializes from drift lines (wrack), so discontinuous drift lines would cause an initially discontinuous or patchy development of dune plants (and therefore discontinuous dunes). Therefore continuous or discontinuous plant initialization (in the alongshore) can control the initial alongshore continuity of the foredune (continuous or hummocky).

Given a discontinuous initial plant configuration, the spaces between plant sites infill through the establishment of new plants, and/or the lateral expansion of existing plants via rhizomes (e.g., Keijsers et al., 2015). In this way, plant dynamics can also control the existence of hummocky dunes. Some plants grow laterally faster than others — Godfrey and coworkers (Godfrey 1977; Godfrey and Godfrey, 1973; Godfrey et al., 1979) found that dunes of the northeastern U.S. had more continuous ridges than the hummocky isolated dunes of the southeastern U.S., which they attributed to differences in plant lateral growth rates for the dominant species in each region.

Geological and geomorphic templates have also been used to explain variability in dune height. Low areas without dunes can remain low because of shell or coarse-grained lags, a high water table that causes plant stress, and/or climatic conditions such as cold temperatures prohibiting plant growth (e.g., Mountney and Russell, 2006; 2009; Wolner et al., 2012; Ruz and Hesp, 2014; Ruz et al., 2017a). Godfrey (1977) hypothesized that barrier island orientation relative to the prevailing winds exerts a control on foredune morphology, with taller dunes occurring when winds blow directly onshore, perpendicular to the shoreline. Sediment supply has also been implicated in causing alongshore dune height variability — specifically that geomorphic and geologic framework influences the morphology of bars, beaches, and sediment supply, therefore controlling the height of coastal dunes (Houser et al 2008; Houser and Mathew, 2011).

These proposed mechanisms may explain the formation of hummocky dunes, though foredunes, once formed, are dynamic features, evolving and growing through time. Both mature hummocky dunes as well as continuous dune ridges may evolve from initially hummocky dunes. Ritchie and Penland (1988a, 1988b, 1990) developed a conceptual model of coastal foredune development following flattening of foredune topography by a storm, stating that a mature, continuous foredune can develop from a washover terrace given sufficient time. The transition from washover terrace (a low surface) to a continuous dune requires individual incipient dunes to grow and merge, eventually developing into a single continuous ridge. (Ritchie and Penland, 1988; 1990; Pye, 1983; Carter and Wilson 1990; Davidson-Arnott and Fisher, 1992; Mathew et al., 2010; Montreuil et al., 2013). Such a conceptual model, consistent with widely observed field conditions, does not address why some initially hummocky foredunes coalesce to a linear foredune ridge, while others remain hummocky, having variable dune height in the alongshore direction, though Godfrey (1977) discussed the potential for recurring storm events to prevent the coalescing of hummocky dunes, even in locations where vegetation grows rapidly in the lateral direction.

In this contribution we develop and explore a model of coastal foredune growth and hummocky dune evolution — that is consistent with this previous work — to better understand the mechanisms behind the development of hummocky foredunes in the alongshore direction. Previous work by Moore et al (2016) has investigated the cross-shore dynamics. Our

work here is a quantitative investigation of several of the hypotheses of Godfrey (1977), notably that vegetation exerts a fundamental control on alongshore dune morphology. Our findings suggest that, given no pre-existing template and sufficient time prior to occurrence of a storm event, alongshore hummocky dunes eventually coalesce to form a continuous coastal foredune ridge. Model results are well explained by a predictive rule for both the coalescing timescale and the height
of the coalesced dune that depend on the initial spatial distribution of dune vegetation (which controls the location of incipient dunes), and the lateral and vertical growth rate of vegetation.

## 2 Ecomorphodynamic Model

We use a recently developed model of coastal dunes that includes the lateral propagation of vegetation (Moore et al., 2016). This model is based on the coastal dune model of Durán and Moore (2013), itself based on previous models used
to study a variety of dunes (e.g., Parteli et al., 2009; Durán and Hermann 2006; Durán et al., 2010). We briefly summarize the model and the vegetation formulation below.

Given an initial topography $h(x, y)$ and a vegetation field, the model computes the bed shear stress perturbation due to the presence of a non-flat topography (Weng et al. 1991), modified by a separation bubble (when there is flow separation; Kroy et al., 2002) and the subsequent shear stress reduction due to vegetation (Raupach et al., 1993). From the bed shear
stress field, the local non-uniform sand flux and sand flux divergence is then computed at every position (Kroy et al 2002; Durán et al 2010) — this determines the temporal change in topography. Sand avalanching occurs down the steepest descent gradient when topography exceeds the angle of repose. After the topography has been updated, the change in the vegetation field is calculated (itself dependent on the local accretion/erosion rate).

We use a simplified version of the vegetation formulation presented in Moore et al. (2016), which is itself a
modification of earlier models (Durán and Moore, 2013, 2015; Durán and Hermann 2006). We now present the simplified vegetation model and then discuss the physical interpretation for the two key sensitivity parameters.

The vegetation is parameterized by the cover fraction $\rho_{veg}$. The growth and propagation of vegetation is modeled by an advection equation of the form:

$$\frac{d\rho_{veg}}{dt} = C|\nabla\rho_{veg}| + G_0\rho_{veg}(1 - \rho_{veg}), \tag{1}$$

where the first term is the lateral propagation of vegetation at rate $C$ due to rhizome growth, and the second term is the local growth of biomass to maximum cover $\rho_{veg} = 1$. The intrinsic growth rate ($G_0$) is assumed to increase with the deposition rate $\max(\frac{dh}{dt}, 0)$ and to vanish near to the shoreline ($x < L_{veg}$, where $x$ is the distance to the shoreline). This is represented by a Heaviside function ($\Theta$) that is unity when distance to the shoreline is sufficient for plant growth $\left((x - L_{veg}) > 0\right)$, and 0 otherwise:


$$G_0 = H_v^{-1} \max\left(\frac{dh}{dt}, 0\right) \Theta(x - L_{veg}),$$ (2)

The lateral vegetation propagation rate $C$ is also assumed to increase with the deposition rate and to vanish for steep slopes ($\tan\theta_c < |\nabla h|$; where $\theta_c$ is 15 degrees and is based on field observations from Moore et al., (2016). This is represented by a Heaviside function ($\Theta$) that is unity when the slope of the land surface is not beyond a threshold $((\tan\theta_c - |\nabla h|) > 0)$ and

0 otherwise :

$$C = \beta \max\left(\frac{dh}{dt}, 0\right) \Theta(\tan\theta_c - |\nabla h|),$$ (3)

This formulation of vegetation growth has two parameters that reflect the sensitivity of plants to changes in surface topography. First, the intrinsic growth rate ($G_0$) of vegetation in the logistic model is sensitive to plant burial, to simulate the

behavior of dune-building plants that are stimulated by burial (e.g., Maun and Perumal, 1999; Maun, 2004; Gilbert and Ripley, 2010). This sensitivity term $H_v$, with dimensions of [L], encodes the efficiency of vertical plant growth after burial. Larger $H_v$ result in smaller values of $G_0$ and therefore slower plant growth, implying that burial is more effective at reducing plant basal area. Second, the lateral propagation of vegetation is sensitive to burial rate and the spatial gradient of cover density. Here, the dimensionless coefficient $\beta$ can be interpreted as the efficiency of rhizome propagation after burial. A

larger $\beta$ results in faster plant propagation from place to place. Note that vertical growth rate relies exclusively on $H_v$, but lateral expansion relies on the spatial gradient of vegetation cover and therefore depends indirectly on $H_v$. If $H_v$ is large, the vertical growth rate is slower and this will cascade to slowness in lateral growth rate (and vice versa).

The model is integrated in a two dimensional grid (64 m alongshore and 100 m cross-shore with 1 m grid size) with periodic alongshore boundary conditions. The shoreline is set to a fixed location and vegetation is 'seeded' in one band at an

identical cross-shore location (40 m from the shoreline). There is a gap in this seeding located near the center of the model domain. The seeded 'line' represents the development of vegetation around a driftline of wrack, and is set at the seaward vegetation limit of plant growth (e.g., Durán and Moore 2013, Kuriyama et al., 2005). As a consequence, vegetation does not propagate seaward in model experiments. We track the evolution of the unplanted gap as a single representative example of an initially unvegetated gap in an alongshore foredune. In the absence of observational data that reveals the degree to which

dune-building vegetation establishes via seed versus lateral propagation, beyond the initial 'seeding' we allow plants to establish in unvegetated cells only by lateral propagation, which can be thought of as encompassing establishment via both mechanisms.

Forcing conditions (i.e., undisturbed shear velocity $U^* = 0.35\ m/s$) are kept constant for all model experiments, but we vary the characteristics of the model vegetation to mimic variability in vertical and lateral plant growth rates.

Experiments are shown for a range of vegetation lateral growth parameter values spanning over an order of magnitude ($10 \geq \beta \geq 0.1$), vertical growth parameter values spanning an order of magnitude ($0.4\ m \geq H_v \geq 0.04\ m$), and unvegetated gap sizes ($10 - 20\ m$).

**3 Results**

From the initial condition, the model domain evolves to fill in the unvegetated gap (Figure 2). Initially the vegetation grows from the planted location in the vertical and lateral direction. Initially planted locations evolve into developed foredunes. Within the unvegetated gap, only minor vertical elevation changes occur prior to the establishment of vegetation (via lateral propagation from the vegetated line). After the establishment of vegetation, the initially unvegetated sites become vegetated and grow vertically into a mature foredune. In the final model state, there is no evidence in the former dune gap to suggest that the site was once unvegetated. All model results yield a consistent maximum dune height of between 3.6 - 3.9 m.

We now focus on the lag in height between the unplanted gap and the surrounding planted dune — we refer to this difference as 'hummockiness', the difference in elevation between the dune under the initially planted area compared to the central location at the initially unvegetated gap. Hummockiness first increases with time as the initially unplanted site lags behind the planted locations in both vegetation cover and vertical elevation. Figure 3 is a partial phase plane for all model results displaying hummockiness plotted against the height at the planted dune site. This partial phase space of the model allows for inspection of the trajectory of model results as they evolve from hummocky dunes to coalesced dunes. Initial trajectories all start at the (0.3, 0) mark (the beach is initially at an elevation of 0.3 m, with 0 hummockiness), and evolve in a clockwise fashion as the initially planted sites grow vertically at a faster rate than the unvegetated gap. After the propagation of vegetation into the initially unvegetated gap, the dune in the gap grows vertically at a rate faster than the vegetated sites (which has slowed in vertical growth as it nears the maximum theoretical dune height). This leads all trajectories toward a hummockiness of 0. Note that no time scale is shown in this phase space.

Two trajectories are shown in Figure 3 to illustrate that the maximum hummockiness (the peak) is a function of $H_v$ and $\beta$. As the lateral vegetation growth parameter ($\beta$) decreases from 10 to 0.1, the lateral growth rate slows down, which increases the variability in alongshore dune crest heights — hummockiness tends to increase (Figure 4A). On the other hand, an increase in the vertical parameter $H_v$ (plants are more sensitive to burial) slows the growth rate of vegetation thereby increasing the maximum hummockiness (Figure 4A). The unvegetated gap width also plays a role in controlling hummockiness as smaller initially unvegetated gap widths result in faster dune coalescing (Figure 4B)

The general behavior of hummockiness and coalescing lends itself to heuristic analysis. Since the development of coastal dunes relies on the feedback between vegetation growth and aeolian sediment transport, maximum hummockiness occurs at the moment just before the center of a given gap transitions from unvegetated to vegetated (at which point the surrounding vegetated dunes have grown for some time). Therefore maximum hummockiness is related to gap size and lateral propagation of plants —which from (2) and (3) depends on $\beta$ and $H_v$ (via the spatial gradient in vegetation cover). For example, small gap size, high $\beta$ (fast lateral growth of vegetation) and low $H_v$ (fast vertical growth of vegetation) lead to low maximum hummockiness and vice versa. Results from all model simulations conform to this general behavior (Figure 4A and 4B).

Gap size, lateral growth rate of vegetation, and vertical plant sensitivity also impact model timescales for the alongshore coalescing of hummocky dunes. Maximum hummockiness occurs later (Figure 5A) and dunes take longer to coalesce (Figure 5B) with decreasing lateral growth rate of vegetation, increasing plant sensitivity to burial, and increasing gap size.

The lateral propagation rate (P) of the dune is defined as the time needed to propagate the crest a given lateral (alongshore) distance—the lateral spreading rate of the dune crest. This rate encompasses the spreading rate of the plant, and the biophysical feedbacks that lead to dune growth. Lateral dune propagation rate is defined as $P = (0.5 \times W)/T_a$ where $(0.5 \times W)$ is the half width of the gap $(W)$ and $T_c$ is the time to coalescing. The half width of the gap is used since all model experiments include unvegetated gaps that fill in from both sides. Within the limits of the model experiments, results are

well described by an equation of the form:

$$P = K_1 \beta + \frac{K_2}{H_v}, \tag{4}$$

where $K_1$ and $K_2$ are dimensional parameters (6.5 m/yr and 1.9 m$^2$/yr). A high $\beta$ (fast lateral growth of vegetation) and low $H_v$ (fast vertical growth of vegetation) lead to fast lateral propagation of the dune crest. Figure 5 shows the modeled vs.

predicted propagation times derived from (4).

     Rewriting equation 4, the coalescing time can be written as:

$$T_c = \frac{W}{K_1 \beta + \frac{K_2}{H_v}} \quad or \quad T_c = \frac{W H_v}{K_1 \beta H_v + K_2}, \tag{5}$$

Following Durán and Moore (2013), we assume in the model a constant wind shear velocity ($U* = 0.35\ m/s$) that

represents typical wind conditions during dune growth. Because in reality conditions sufficient for transport do not occur all the time, Durán and Moore (2013) suggest that model time can be converted to real time by multiplying model time by a factor ($r_t$) that varies from 0 to 1 and represents the fraction of time there is no transport. Therefore reduction in the flux of sand from beach to dune, because of low wind speeds, large grain sizes, or narrow beaches, can be encapsulated through variation in $r_t$ and has an effect similar to decreasing $\beta$ and increasing $H_v$.

The height of the dune crest at the moment of coalescing ($H_c$) can be described by:

$$H_c = H_{max} \left( 1 - e^{\left( \frac{-T_c}{T_f} \right)} \right) + Z, \tag{6}$$

where $H_{max}$ is maximum dune size, $T_f$ is formation time of the planted sites, and $Z$ is the initial beach elevation at the site of dune nucleation (here 0.3). Both $H_{max}$ and $T_f$ are functions of the seaward vegetation growth limit as well as other relevant parameters, defined in Durán and Moore (2013). Figure 6 is the modeled vs. predicted dune height at coalescing calculated

from (6).

**4 Discussion and Implications**

Godfrey (1977) and Godfrey et al. (1979) observed that foredunes change from irregular, 'hummocky' dunes in the southeastern U.S. to contiguous long-crested dunes in the northeastern U.S. This change in observed dune morphology is attributed to changes in foredune species dominance (Godfrey and Godfrey, 1973; van der Valk, 1975; Woodhouse et al.,

1977, Godfrey, 1977; Godrey et al., 1979). From Virginia northward, foredunes are dominated by *Ammophila breviligulata* (American Beachgrass) while south of Virginia, *Uniola paniculata* (Sea Oats) dominates foredunes (Wagner, 1964; Godfrey, 1977; Duncan and Duncan, 1987; Lonard et al., 2011). On the east coast, *A. breviligulata* and *U. paniculata* exhibit similar rates of vertical growth (including the adapted response of increasing growth rates when buried by moderate amounts of sand; Disraeli, 1984; Maun, 2004, Ehrenfeld 1990, Lonard, et al. 2011; Wagner, 1964). However *A. breviligulata* and *U.*

*paniculata* exhibit differences in rates of lateral growth, 1-3 m/yr and 0.6-1 m/yr respectively (Woodhouse et al., 1977; Ehrenfeld, 1990 Lonard et al., 2011). The slower lateral growth rate of *U. paniculata* provides a potential explanation for the observation of hummocky dunes along the southeastern U.S. coast. This species-specific control on dune morphology likely arises from differences in growth form, similar to observations that explain species-specific dune morphology along the U.S. West Coast (Hacker et al., 2012; Zarnetske et al., 2012). We can understand these differences in the context of model

findings– though *A. breviligulata* and *U. paniculata* may have similar vertical growth characteristics ($H_v$ is identical), their lateral growth rates (encoded here as $\beta$) are different, resulting in differences in dune hummockiness (Figure 4A) and coalescing time (Figure 5b). The dominant dune-building plant of the southeastern U.S. has a slower lateral growth rate and therefore a longer coalescing time, likely leading to the increased prevalence of hummocky foredunes in this region. Evidence that even *U. paniculata* can form continuous dune ridges is present on Sapelo Island, Georgia, U.S.  The lack of a

major hurricane strike in this region (Bossak et al., 2014) is manifest in the continuous ridge topography even though the foredune is dominated by *U. paniculata* (Monge and Stallins, 2016; Stallins 2005; Stallins and Parker, 2003).

However, the numerical finding that hummocky dunes always coalesce if given sufficient time suggests that differences in species-specific lateral growth rates alone are not sufficient to explain hummockiness that persists through time.  A more complete explanation likely comes from combining our finding that coalescing time lengthens with decreasing

lateral growth rate of the dominant dune-building grass, with the suggestion by several studies that low areas (and therefore hummocks) are maintained by overwash during high water events (Godfrey, 1977; Hosier and Cleary, 1977; Ritchie and Penland, 1988). We can understand this using (7)—if the recurrence time for high water events ($R$) is shorter than the coalescing time $T_c$, existing hummockiness will likely be maintained because low areas are more likely to be overwashed than adjacent higher dunes on either side.  When this occurs, the dune-building process in the low areas is reset, increasing

hummockiness until vegetation again becomes established in the overwashed zone. Conversely, if $R \gg T_c$, hummockiness will tend to decrease through time because there will be sufficient time between storms for coalescing to occur.  Along the southeast U.S. coast it appears that $R < T_c$ given the previous observations that hummocky dunes are prevalent there and the slow lateral growth rate of *U. paniculata,* Thus, although hummockiness appears to be an intrinsic feature of foredunes along

the southeast coast of the U.S., model results suggest that hummockiness is actually a transient characteristic of foredunes that only becomes persistent when coalescing time is slow relative to the frequency of storms capable of resetting the dune-building process in the the low areas between hummocks.

In the case of $R > T_c$, environmental conditions may be conducive to bistable dynamics in the alongshore direction—similar to the cross-shore models of Durán Vinent and Moore (2015) and Goldstein and Moore (2016)—with alternating stretches of dunes near the maximum height and lower intervening areas. In addition to storms, other factors such as a high water table, low sediment supply, grain size variability, development of shell lag, and climatic conditions may also result in suppression of the coalescing of coastal foredunes (Mountney and Russell, 2006; 2009; Wolner et al., 2012; Hoonhout and de Vries, 2016; Ruz and Hesp, 2014; Ruz et al., 2017a). Feedbacks between the wind, dune vegetation and sediment transport that are specific to hummocky dunes may also alter the rates of coalescing (Barrineau and Ellis, 2013; Gilles et al., 2014), such as the development of high wind velocity regions located adjacent to hummocky dune forms (Hesp and Smyth, 2017). Work here does not address observations of older foredune ridges that lose their continuous morphology as a result of plant succession, erosion via rain and flow in rivulets, or trampling (Levin et al 2009; 2017). Additionally the potential for lag between 'fast' cross-shore beach recovery time vs. slower cross-shore vegetation recovery time (e.g., Castelle et al 2016; Keijsers et al., 2016; Ruz et al., 2017b) could introduce novel dynamics that are not explored in this work.

There exists a potential for climate change to alter the range of the two dominant species of dune-building grasses along the U.S. East Coast. Plantings of *A. breviligulata* south of VA tend to die as a result of blight, pests, drought intolerance, and intolerance of high temperature (Seneca, 1972; Singer et al., 1973; van der Valk, 1975; Woodhouse et al., 1977; Odum et al., 1987, Seliskar and Huettel, 1993). A warming climate might lead to further northward expansion of *U. paniculata*, which is currently restricted in northward extent by temperature (Seneca, 1972; Godfrey, 1977)—northern expansion of the range has already been observed (Zinnert et al., 2011; Stalter and Lamont, 1990; 2000) and is being sought in selective breeding trials (USDA, 2013). Additionally, glasshouse experiments have reported that *A. breviligulata* is negatively impacted by competition with *U. paniculata* (Harris et al. 2017; Brown et al., 2017). Because changes in $\beta$ between these two dune-buiding species affects variability in alongshore dune height, a change in the dominant dune-building species from *A. breviligulata* to *U. paniculata* has the potential to decrease the protection provided by dunes during high water events. Changes in storminess may also impact the hummockiness of coastal foredunes, with an increase in storm intensity or frequency leading to a greater tendency for dunes to be hummocky and therefore to provide less protection to habitats behind them. Here, we have focused on the development of hummocky dunes from an initially flat condition, but Lazarus and Armstrong (2014) discuss the potential for storm events to create regularly spaced overwash throats (via self-organization) that could also set up hummocky dune topography. Although beyond the scope of this effort, observational work aimed at assessing the relationships among storm frequency/magnitude, species composition of dune-building vegetation and dune development (e.g., van Puijenbroek et al., 2017a; 2017b) will be useful in addressing the future implications of model results presented here as climate change is anticipated to alter each of these factors.

**Acknowledgements**

EBG thanks Theo Jass and Elsemarie deVries for valuable discussions regarding this work. We thank 2 anonymous reviewers for comments on the manuscript. Funding was provided by NSF-GLD (EAR-1324973) and the Virginia Coast Reserve Long-Term Ecological Research Program (NSF DEB-123773).

5   **Appendix A: Variables**

| Symbol | Variable Name |
|---|---|
| $h$ | Elevation |
| t | Time |
| $\rho_{veg}$ | Vegetation cover fraction |
| C | Lateral vegetation propagation rate |
| $G_0$ | Intrinsic growth rate |
| $L_{veg}$ | Seaward limit of vegetation growth |
| $\theta_c$ | Critical topographic angle where vegetation stops expanding laterally |
| $H_v$ | Vertical vegetation growth sensitivity term |
| $\beta$ | Lateral vegetation growth sensitivity term |
| W | Half width of unvegetated gap (i.e., half width of plant spacing) |
| P | Lateral propagation rate of dune |
| $T_c$ | Time to coalescing |
| $K_1$ | Dimensional parameter |
| $K_2$ | Dimensional parameter |
| $H_{max}$ | Maximum dune size |
| $T_f$ | Dune formation time at planted sites (time to $H_{max}$) |
| Z | Initial beach elevation at site of dunes |
| R | Recurrence time for high water events |

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

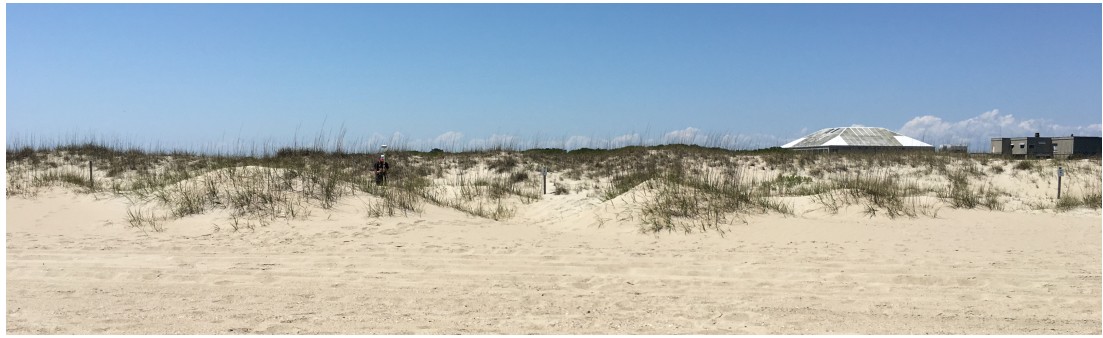

**Figure 1: Ground-based photo of 1-2 m hummocky foredunes covered with *Uniola paniculata* at Fort Fisher State Recreation Area, NC, USA. (Note the person in the center left wearing black with a 2 m fixed height survey pole for scale). The hummocky foredunes are seaward of an older continuous dune ridge.**

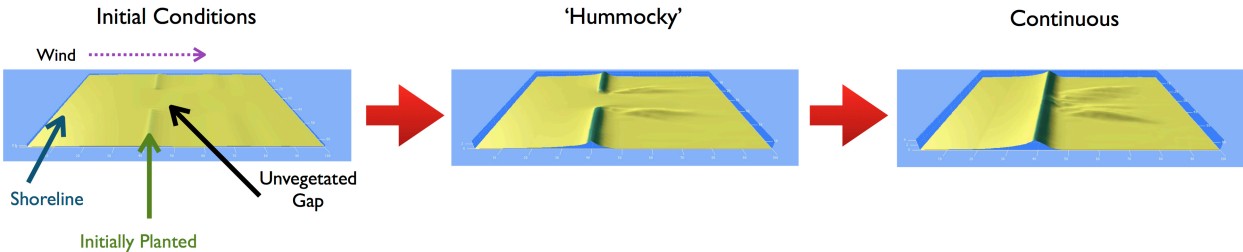

**Figure 2: Numerical model definition sketch. A) Initial conditions; B) Formation of foredune and the infilling of the initially unvegetated gap. C) The final continuous foredune ridge at the maximum theoretical dune height.**

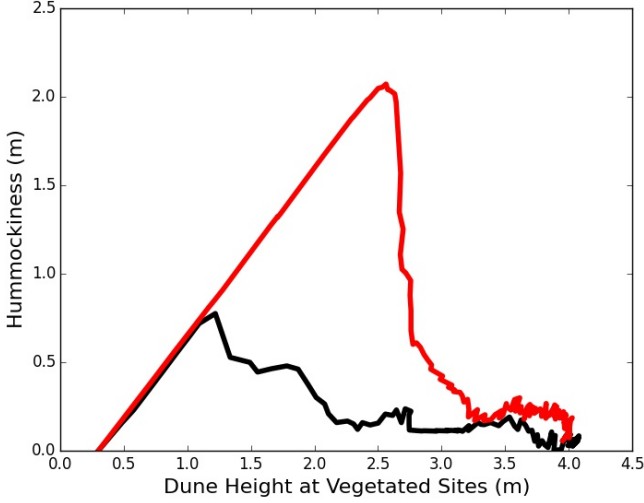

10    **Figure 3: Phase plot of two numerical model experiments — the experiment in black has larger growth parameter ($\beta$) and therefore faster lateral growth and lower hummockiness than the red experiment. All model iterations begin at (0.3, 0), reflecting**

the initial height of the planar sloping surface (0.3 m) at the location of the dune vegetation plantings. As the model iterates, the hummocky dunes develop, as vegetated sites grow in height more than unvegetated sites (which must wait for vegetation to grow before increasing in height). After vegetation propagates to these sites, a continuous foredune ridge develops and hummockiness reduces to zero. The maximum hummockiness and the trajectory through phase space is set by gap size (w), vertical vegetation growth parameter (Hveg) and lateral vegetation growth parameter (**β**).

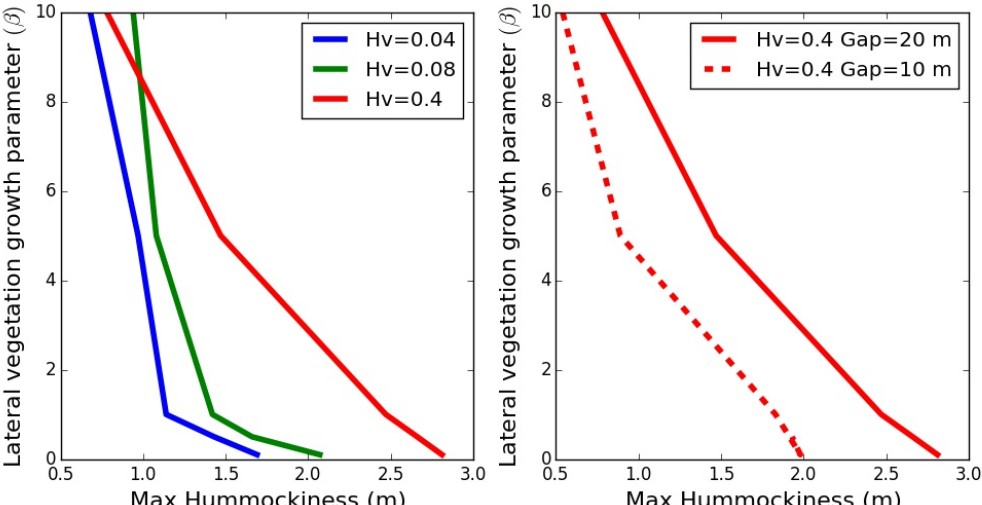

**Figure 4: a) Maximum hummockiness (m) as a function of $H_v$ (vertical vegetation growth parameter) and $\beta$ (lateral vegetation growth parameter); B) Maximum hummockiness (m) as a function of $\beta$ and unvegetated dune gap size**

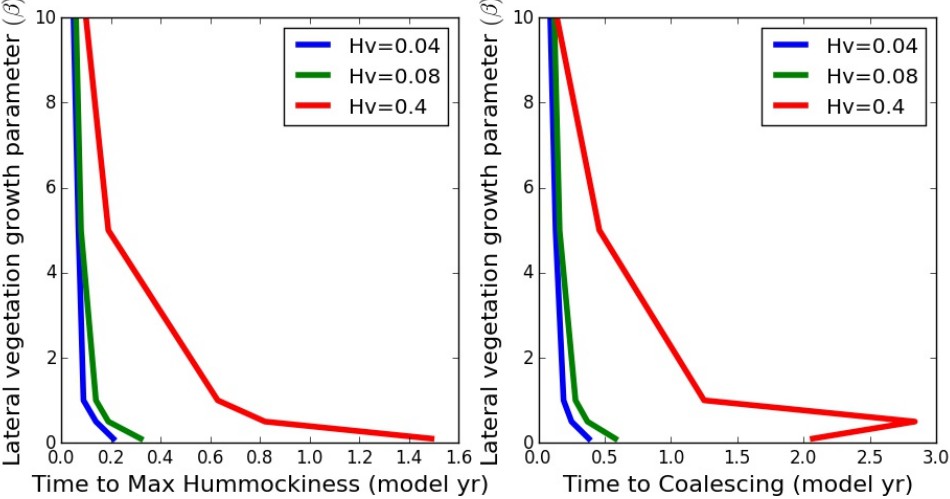

**Figure 5: The impact of changes in the vertical vegetation growth parameter ($H_v$) and lateral vegetation growth parameter ($\beta$) on a) the time of maximum hummockiness; and b) the time when coalescing occurs.**

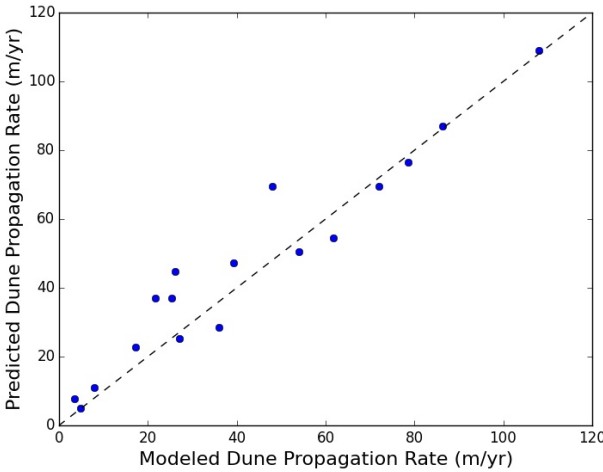

**Figure 6: Modeled lateral dune propagation rate vs. predicted propagation rate from (6). Black line is 1:1.**

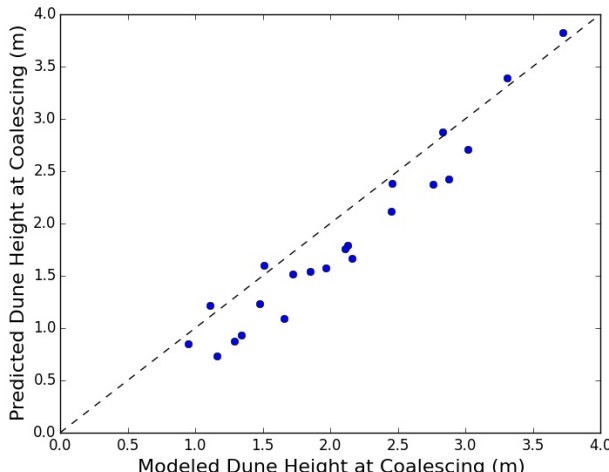

**Figure 7: Modeled dune elevation at coalescing vs. predicted dune elevation at coalescing from (8). Black line is 1:1.**

