# Peer review of "Lateral vegetation growth rates exert control on coastal foredune "hummockiness" and coalescing time"

_Earth Surface Dynamics, 2017_

## Referee Comment (RC1) · Anonymous Referee #1 · 21 Mar 2017

General comments

This paper makes use of a model to examine the way growth of vegetation leads to development of a continuous foredune ridge. The paper could use a better balance in its presentation of the need for the model and its application to resolve the research issues. The research issues where the model can potentially be of use are now introduced in better detail in the discussion and implications section. Godfrey has already explained much of the reason for hummockiness. His work should be presented in detail up front and discussed in terms of modern needs to demonstrate that there is a need to subject his insightful conceptual model to a test. The discussion of the mechanics of the model occupies most of the paper. The results section is actually an extension of the methods because it tell how the model is to be used rather than what use of the model tells us.

[Figure]

The conclusion that given sufficient time and lack of external forcing, hummocky dunes can form dune ridges seems self evident if the model is designed to get to this stage. What are the practical implications of this statement? Is there evidence that this kind of end stage can be achieved in nature, especially for species that now tend to form hummocky dunes and in light of sea level rise and potential increase in storminess? Even Ammophila-dominated dunes may tend toward hummockiness rather than a linear form, with an increase in frequency/magnitude of storms. This concept is introduced at the end of the paper but not used to determine the applicability of the model or its use.

The word "annealing" is not intuitively obvious from a standard definition of the term. In any case, the word should be eliminated from the title, where it cannot be defined in the context used here. The title should be reworded in any case. The paper is not actually about vegetation controls, which would involve a much more comprehensive discussion of growth patterns, rates, etc. related to specific vegetation types. The title should reflect the use of the model, if the model remains the primary focus of the paper.

Specific comments

Abstract The first sentence is misleading because this paper is not about building dunes for shore protection. The implication is that hummockiness is a bad thing, when it may represent a balanced geomorphic-ecologic condition. The goal expressed on lines 14 and 15 should be more specific to the paper because the causes and dynamics of hummocky foredunes have already been examined in terms of vegetation characteristics. Lines 18-19: Why not state the predictive rule right in the abstract and specifically identify the two parameters that control lateral and vertical vegetation growth? The findings and explanation for the findings identified in Lines 20-23 are already documented in the literature. More original findings of this study should be identified.

First paragraph of the introduction: Coastal dunes can be initiated without colonizing

plants. It may be useful to identify the starting condition for incipient dune formation (e.g. overwash), or is the discussion about new dunes forming seaward of an existing foredune?

Discussion of the cross-shore component would be important in the shore protection context. The word "hummocky" is introduced, and apparently evaluated (Fig. 2) as a two-dimensional concept, but it has cross-shore expression as well. A sentence or two dismissing or assuming away the cross-shore aspects should be inserted, but that may require eliminating the shore-protection context as well because volume is critical to shore protection.

First complete paragraph page 2 (beginning Line 8). This would be a good place to introduce the Godfrey model in greater detail and identify how this paper will expand or refine it.

Lines 26-28 on Page 2: This is not an open question, which is why the Godfrey model should be introduced in sufficient detail to identify what the remaining open question is and how the model can answer it.

Last paragraph of paper: I suggest eliminating this paragraph because it implies that the model is not ready for use.

---

## Referee Comment (RC2) · Anonymous Referee #2 · 24 Apr 2017

The authors build upon their previously developed numerical ecomorphodynamic models, to demonstrate some of the factors controlling the hummockiness of a foredune, at the absence of external forcing. This is a good paper, and it would be very interesting to see the model compared to field and remote sensing observations of foredunes, so as to provide some validation to the model.

Following are some more detailed comments:

The term "annealing" may be mistakenly interpreted as if a foredune is annealed and washed by waves, whereas the authors mean that the hummockiness is annealed, not the foredune. I suggest that the authors use a different term throughout the paper.

Add a table showing all variables, abbreviations and their meaning, to make it easier for the readers to follow the equations which are developed

p. 2. l. 21: Continuous dune ridges may also become less continuous and hummocky with time, see: Levin, N., Tsoar, H., Herrmann, H. J., Maia, L. P., & Claudino-Sales, V. (2009). Modelling the formation of residual dune ridges behind barchan dunes in North‐east Brazil. Sedimentology, 56(6), 1623-1641.

p.4 l. 12-13: Is it a reasonable assumption, that plants establish "only by lateral propagation"?

Figures 4, 5: State in the figure captions what does Hv represent.

Discussion: While hummocky foredunes may indeed anneal to form continuous foredunes at their early life stages, later on, foredunes often "lose" their continuous form, as large shrubs and trees start to develop, and additional process of erosion take place. See Figure 8 in Levin, N., Jablon, P. E., Phinn, S., & Collins, K. (2017). Coastal dune activity and foredune formation on Moreton Island, Australia, 1944–2015. Aeolian Research, 25, 107-121. I also refer the authors to Castellte et al. (2017), who show that following a storm, foredune vegetation recovery time may be much longer than sand volume recovery time: Castelle, B., Bujan, S., Ferreira, S., & Dodet, G. (2017). Foredune morphological changes and beach recovery from the extreme 2013/2014 winter at a high-energy sandy coast. Marine Geology, 385, 41-55.

---

## Editor Comment (EC1) · A.C.W. Baas (Editor) · 12 May 2017

The manuscript has received two review reports that the authors are asked to respond to now.

While R2 is overall supportive of the work the main caution they raise is that various studies show that the annealing of hummocky foredunes into a continuous dune ridge is not the inevitable outcome that this work appears to present it to be, and that many examples exist of hummocky foredunes that do not anneal, or continuous dune-ridges that break apart again. The paper should therefore be revised to recognize these alternative states and landscape trajectories and to evaluate the modelling work within that context.

This overall concern is echoed by R1, who also raises a number of other, more critical

and detailed concerns about the manuscript in its present form and has suggested a number of improvements to the background and the discussion of this work.

In your response to referees and your revision of the manuscript I therefore ask that you address the concerns and issues raised by R1 in particular.

---

## Author Comment (AC2) · 12 Jun 2017

**Response to Reviewer 1**

Reviewer comments in plain text
**Author comments are in BOLD**
ESurfD Manuscript text is in *italics*
**Added Text is in *Bold Italics***

**We thank Referee #1 for taking the time to read and review our manuscript. We address each specific comment below:**

General Comments:
This paper makes use of a model to examine the way growth of vegetation leads to development of a continuous foredune ridge. The paper could use a better balance in its presentation of the need for the model and its application to resolve the research issues. The research issues where the model can potentially be of use are now introduced in better detail in the discussion and implications section. Godfrey has already explained much of the reason for hummockiness. His work should be presented in detail up front and discussed in terms of modern needs to demonstrate that there is a need to subject his insightful conceptual model to a test. The discussion of the mechanics of the model occupies most of the paper. The results section is actually an extension of the methods because it tell how the model is to be used rather than what use of the model tells us.

**We have now clarified the text of the paper to explain how our focus is on quantitative, predictive rules compared to the observational work that was done in the 1970s.**

**In the Abstract (Page 1 Line 15-21):**
*"**Model results yield a** predictive rule for the timescale **of coalescing and the height of the coalesced dune** that depends on initial plant dispersal and two parameters that control the lateral and vertical growth of vegetation, respectively. Our findings **agree with previous observational and conceptual work** — whether or not hummockiness will be maintained depends on the time scale of **coalescing** relative to the recurrence interval of high water events that reset dune-building in low areas between hummocks. **Additionally, our model reproduces the observed** tendency for foredunes to be hummocky along the southeast coast of the U.S. where lateral vegetation growth rates, and thus **coalescing** times, are likely longer. "*

**Page 2 Line 32 to Page 3 Line 6**
*"In this contribution we develop and explore a model of coastal foredune growth and hummocky dune evolution —that is consistent with this previous work — to better understand the mechanisms behind the development of hummocky **foredunes in the alongshore direction. Previous work by Moore et al (2016) has investigated the cross-shore dynamics. Our work here is a quantitative investigation of several of the hypotheses of Godfrey (1977), notably that vegetation exerts a fundamental control on alongshore dune morphology.** Our findings suggest that, given no pre-existing template and sufficient time prior to occurrence of a storm event, **alongshore** hummocky dunes eventually **coalesce** to form a continuous coastal foredune ridge. **Model results are well explained by a predictive rule for both the coalescing timescale and the height of the coalesced dune that depend on the** initial spatial distribution of dune vegetation*

*(which controls the location of incipient dunes), and the lateral and vertical growth rate of vegetation."*

**We also include additional references to Godfrey in the introduction:**

**Page 2 Line 13-18:**
*"Geological and geomorphic templates have also been used to explain variability in dune height. Low areas without dunes can remain low because of shell or coarse-grained lags, a high water table that causes plant stress, **and/or climatic conditions such as cold temperatures prohibiting plant growth** (e.g., Mountney and Russell, 2006; 2009; Wolner et al., 2012; Ruz and Hesp, 2014; Ruz et al., 2017a). **Godfrey (1977) hypothesized that barrier island orientation relative to the prevailing winds exerts a control on foredune morphology, with taller dunes occuring when winds blow directly onshore, perpendicular to the shoreline**."*

**Page 2 Line 23-31**
*"Ritchie and Penland (1988a, 1988b, 1990) developed a conceptual model of coastal foredune development following flattening of foredune topography by a storm, stating that a mature, continuous foredune can develop from a washover terrace given sufficient time. The transition from washover terrace (a low surface) to a continuous dune requires individual incipient dunes to grow and merge, eventually developing into a single continuous ridge. (Ritchie and Penland, 1988; 1990; Pye, 1983; Carter and Wilson 1990; Davidson-Arnott and Fisher, 1992; Mathew et al., 2010; Montreuil et al., 2013). Such a conceptual model, consistent with widely observed field conditions, **does not address** why some initially hummocky foredunes coalesce to a linear foredune ridge, while others remain hummocky, having variable dune height in the alongshore direction, **though Godfrey (1977) discussed the potential for recurring storm events to prevent the coalescing of hummocky dunes, even in locations where vegetation grows rapidly in the lateral direction."***

*Page 3 Line 1-2:*
*"**Our work here is a quantitative investigation of several of the hypotheses of Godfrey (1977), notably that vegetation exerts a fundamental control on alongshore dune morphology**."*

**We also add a citation to another Godfrey paper in the manuscript (Godfrey et al 1979)**

The conclusion that given sufficient time and lack of external forcing, hummocky dunes can form dune ridges seems self evident if the model is designed to get to this stage. What are the practical implications of this statement?

**The key result is the quantitative relationship, which is different that the conceptual work of Godfrey. We have modified our introduction to state this more clearly.**

**Page 2 Line 32 to Page 3 Line 6:**

*"In this contribution we develop and explore a model of coastal foredune growth and hummocky dune evolution —that is consistent with this previous work — to better understand the mechanisms behind the development of hummocky **foredunes in the alongshore direction.***

*Previous work by Moore et al (2016) has investigated the cross-shore dynamics. Our work here is a quantitative investigation of several of the hypotheses of Godfrey (1977), notably that vegetation exerts a fundamental control on alongshore dune morphology.* Our findings suggest that, given no pre-existing template and sufficient time prior to occurrence of a storm event, **alongshore** hummocky dunes eventually **coalesce** to form a continuous coastal foredune ridge. *Model results are well explained by a predictive rule for both the coalescing timescale and the height of the coalesced dune that depend on the* initial spatial distribution of dune vegetation (which controls the location of incipient dunes), and the lateral and vertical growth rate of vegetation."

**There is also a line in the abstract that discusses the predictive rule in our abstract:**

**In the Abstract (Page 1 Line 15-21):**
*"**Model results yield a** predictive rule for the timescale **of coalescing and the height of the coalesced dune** that depends on initial plant dispersal and two parameters that control the lateral and vertical growth of vegetation, respectively. Our findings **agree with previous observational and conceptual work** — whether or not hummockiness will be maintained depends on the time scale of **coalescing** relative to the recurrence interval of high water events that reset dune-building in low areas between hummocks. **Additionally, our model reproduces the observed** tendency for foredunes to be hummocky along the southeast coast of the U.S. where lateral vegetation growth rates, and thus **coalescing** times, are likely longer. "*

Is there evidence that this kind of end stage can be achieved in nature, especially for species that now tend to form hummocky dunes and in light of sea level rise and potential increase in storminess?

**This is a good point. We now include several sentences regarding Sapelo Island, GA, USA as an example of a location where continuous dune ridges form even when the species that tend to form hummocky dunes are dominant.**

**Page 7 Line 17-21**
*"The dominant dune-building plant of the southeastern U.S. has a slower lateral growth rate and therefore a longer coalescing time, likely leading to the increased prevalence of hummocky foredunes in this region. **Evidence that even U. paniculata can form continuous dune ridges is present on Sapelo Island, Georgia, U.S.  The lack of a major hurricane strike in this region (Bossak et al., 2014) is manifest in the continuous ridge topography even though the foredune is dominated by U. paniculata (Monge and Stallins, 2016; Stallins 2005; Stallins and Parker, 2003).**"*

Even Ammophila-dominated dunes may tend toward hummockiness rather than a linear form, with an increase in frequency/magnitude of storms. This concept is introduced at the end of the paper but not used to determine the applicability of the model or its use.

**We now discuss storm frequency in the introduction.**

**Page 2 Line 23-31**

*"Ritchie and Penland (1988a, 1988b, 1990) developed a conceptual model of coastal foredune development following flattening of foredune topography by a storm, stating that a mature, continuous foredune can develop from a washover terrace given sufficient time. The transition from washover terrace (a low surface) to a continuous dune requires individual incipient dunes to grow and merge, eventually developing into a single continuous ridge. (Ritchie and Penland, 1988; 1990; Pye, 1983; Carter and Wilson 1990; Davidson-Arnott and Fisher, 1992; Mathew et al., 2010; Montreuil et al., 2013). Such a conceptual model, consistent with widely observed field conditions, does not address why some initially hummocky foredunes coalesce to a linear foredune ridge, while others remain hummocky, having variable dune height in the alongshore direction, though Godfrey (1977) discussed the potential for recurring storm events to prevent the coalescing of hummocky dunes, even in locations where vegetation grows rapidly in the lateral direction."*

**Although such an assessment is beyond the scope of this project, we have also added the following sentence at the end of the manuscript:**

**Page 8 Line 31-34:**

***"Although beyond the scope of this effort, observational work aimed at assessing the relationships among storm frequency/magnitude, species composition of dune-building vegetation and dune development (e.g., van Puijenbroek et al., 2017a; 2017b) will be useful in addressing the future implications of model results presented here as climate change is anticipated to alter each of these factors."***

The word "annealing" is not intuitively obvious from a standard definition of the term. In any case, the word should be eliminated from the title, where it cannot be defined in the context used here.

**We have replaced all uses of the term 'anneal' (referring to closing of the gap between dunes), and its variants, with 'coalesce' (referring to merging of the dunes themselves, and therefore closing of the gap).**

The title should be reworded in any case. The paper is not actually about vegetation controls, which would involve a much more comprehensive discussion of growth patterns, rates, etc. related to specific vegetation types. The title should reflect the use of the model, if the model remains the primary focus of the paper.

**We have also changed the title of the manuscript:**

***Lateral vegetation growth rates exert control on coastal foredune "hummockiness" and coalescing time***

Specific comments:
Abstract The first sentence is misleading because this paper is not about building dunes for shore protection. The implication is that hummockiness is a bad thing, when it may represent a balanced geomorphic-ecologic condition.

**We have rewritten the first lines of the abstract to conform more to the paper:**

**Page 1 Line 10-11**
"*Coastal foredunes form along sandy, low-sloped coastlines and range in shape from continuous dune ridges to hummocky features, which are characterized by alongshore-variable dune crest elevations.*"

The goal expressed on lines 14 and 15 should be more specific to the paper because the causes and dynamics of hummocky foredunes have already been examined in terms of vegetation characteristics. Lines 18-19: Why not state the predictive rule right in the abstract and specifically identify the two parameters that control lateral and vertical vegetation growth? The findings and explanation for the findings identified in Lines 20-23 are already documented in the literature. More original findings of this study should be identified.

**We have rewritten the abstract to better explain that our work provides quantitative backing to the observation and conceptual work that has been done previously.**

**In the Abstract (Page 1 Line 15-21):**
"*Model results yield a predictive rule for the timescale **of coalescing and the height of the coalesced dune** that depends on initial plant dispersal and two parameters that control the lateral and vertical growth of vegetation, respectively. Our findings **agree with previous observational and conceptual work** — whether or not hummockiness will be maintained depends on the time scale of **coalescing** relative to the recurrence interval of high water events that reset dune-building in low areas between hummocks. **Additionally, our model reproduces the observed** tendency for foredunes to be hummocky along the southeast coast of the U.S. where lateral vegetation growth rates, and thus **coalescing** times, are likely longer.* "

First paragraph of the introduction: Coastal dunes can be initiated without colonizing plants. It may be useful to identify the starting condition for incipient dune formation (e.g. overwash), or is the discussion about new dunes forming seaward of an existing foredune?

**We agree that dunes can form without colonizing plant, that is why we use the word 'can be initiated' not 'must be initiated'. We now modify the text to account for incipient dune formation after high water events:**

**Page 1 Line 29-30**

"*New coastal dunes can be initiated when there is sufficient cross-shore width seaward of the existing foredune for plants to colonize (e.g., Hesp, 2002), **or when elevated water levels destroy existing dunes**.*"

Discussion of the cross-shore component would be important in the shore protection context. The word "hummocky" is introduced, and apparently evaluated (Fig. 2) as a two-dimensional concept, but it has cross-shore expression as well. A sentence or two dismissing or assuming

away the cross-shore aspects should be inserted, but that may require eliminating the shore-protection context as well because volume is critical to shore protection.

**Thank you for pointing this out. The cross shore hummockiness was the focus of previous work (Moore et al 2016). We now highlight this at the end of the introduction:**

**Page 2 Line 32 -35:**

*"In this contribution we develop and explore a model of coastal foredune growth and hummocky dune evolution —that is consistent with this previous work — to better understand the mechanisms behind the development of hummocky **foredunes in the alongshore direction. Previous work by Moore et al (2016) has investigated the cross-shore dynamics.***"*

**The shore protection lines in the abstract have been removed**

First complete paragraph page 2 (beginning Line 8). This would be a good place to introduce the Godfrey model in greater detail and identify how this paper will expand or refine it.

Lines 26-28 on Page 2: This is not an open question, which is why the Godfrey model should be introduced in sufficient detail to identify what the remaining open question is and how the model can answer it.

**As discussed earlier, we have added several more references to Godfrey in the introduction, and explicitly discuss how this work relates to the work of Godfrey. We repeat the edits here to answer these two points:**

**Page 2 Line 32 to Page 3 Line 6**
*"In this contribution we develop and explore a model of coastal foredune growth and hummocky dune evolution —that is consistent with this previous work — to better understand the mechanisms behind the development of hummocky **foredunes in the alongshore direction. Previous work by Moore et al (2016) has investigated the cross-shore dynamics. Our work here is a quantitative investigation of several of the hypotheses of Godfrey (1977), notably that vegetation exerts a fundamental control on alongshore dune morphology.** Our findings suggest that, given no pre-existing template and sufficient time prior to occurrence of a storm event, **alongshore** hummocky dunes eventually **coalesce** to form a continuous coastal foredune ridge. **Model results are well explained by a predictive rule for both the coalescing timescale and the height of the coalesced dune that depend on the** initial spatial distribution of dune vegetation (which controls the location of incipient dunes), and the lateral and vertical growth rate of vegetation."*

**We also include additional references to Godfrey in the introduction:**

**Page 2 Line 13-18:**
*"Geological and geomorphic templates have also been used to explain variability in dune height. Low areas without dunes can remain low because of shell or coarse-grained lags, a high water table that causes plant stress, **and/or climatic conditions such as cold temperatures prohibiting***

*plant growth (e.g., Mountney and Russell, 2006; 2009; Wolner et al., 2012; Ruz and Hesp, 2014; Ruz et al., 2017a).* **Godfrey (1977) hypothesized that barrier island orientation relative to the prevailing winds exerts a control on foredune morphology, with taller dunes occuring when winds blow directly onshore, perpendicular to the shoreline**.*"*

**Page 2 Line 23-31**
*"Ritchie and Penland (1988a, 1988b, 1990) developed a conceptual model of coastal foredune development following flattening of foredune topography by a storm, stating that a mature, continuous foredune can develop from a washover terrace given sufficient time. The transition from washover terrace (a low surface) to a continuous dune requires individual incipient dunes to grow and merge, eventually developing into a single continuous ridge. (Ritchie and Penland, 1988; 1990; Pye, 1983; Carter and Wilson 1990; Davidson-Arnott and Fisher, 1992; Mathew et al., 2010; Montreuil et al., 2013). Such a conceptual model, consistent with widely observed field conditions,* **does not address** *why some initially hummocky foredunes coalesce to a linear foredune ridge, while others remain hummocky, having variable dune height in the alongshore direction,* **though Godfrey (1977) discussed the potential for recurring storm events to prevent the coalescing of hummocky dunes, even in locations where vegetation grows rapidly in the lateral direction."**

*Page 3 Line 1-2:*
*"***Our work here is a quantitative investigation of several of the hypotheses of Godfrey (1977), notably that vegetation exerts a fundamental control on alongshore dune morphology***."*

**We also add a citation to another Godfrey paper in the manuscript (Godfrey et al 1979)**

Last paragraph of paper: I suggest eliminating this paragraph because it implies that the model is not ready for use.

**To avoid this incorrect implication we have removed this paragraph and replaced it with the following sentence, which is meant to address how model findings can be tested in the field to yield further insight into potential implications:**

**Page 8; line 31-35:**

"**Although beyond the scope of this effort, observational work aimed at assessing the relationships among storm frequency/magnitude, species composition of dune-building vegetation and dune development (e.g., van Puijenbroek et al., 2017a; 2017b) will be useful in addressing the future implications of model results presented here as climate change is anticipated to alter each of these factors.** "

---

## Author Comment (AC3) · 12 Jun 2017

**Author comments are in BOLD**
**ESurfD Manuscript text is in** *italics*
**Added Text is in** ***Bold Italics***

**A flurry of recent foredune papers was published since our discussion paper was put online, and we have added in text citations and references for several of them. In addition we were alerted to several other papers by emails from colleagues and fellow researchers (after they had seen our discussion paper) — one benefit of the 'Open Review' process.**

**Below are the additions:**

**In the Introduction, page 2; line 13-16:**

*"Geological and geomorphic templates have also been used to explain variability in dune height. Low areas without dunes can remain low because of shell or coarse-grained lags, a high water table that causes plant stress, and/or climatic conditions such as cold temperatures prohibiting plant growth (e.g., Mountney and Russell, 2006; 2009; Wolner et al., 2012;* ***Ruz and Hesp, 2014; Ruz et al., 2017a).****"*

**In the Discussion, Page 8 line 6-16:**

*"In addition to storms, other factors such as a high water table, low sediment supply, grain size variability, development of shell lag, and climatic conditions may also result in suppression of the coalescing of coastal foredunes (Mountney and Russell, 2006; 2009; Wolner et al., 2012; Hoonhout and de Vries, 2016;* ***Ruz and Hesp, 2014; Ruz et al., 2017a****). Feedbacks between the wind, dune vegetation and sediment transport that are specific to hummocky dunes may also alter the rates of coalescing (Barrineau and Ellis, 2013; Gilles et al., 2014),* ***such as the development of high wind velocity regions located adjacent to hummocky dune forms (Hesp and Smyth, 2017). Work here does not address observations of older foredune ridges that lose their continuous morphology as a result of plant succession, erosion via rain and flow in rivulets, or trampling (Levin et al 2009; 2017). Additionally the potential for lag between 'fast' cross-shore beach recovery time vs. slower cross-shore vegetation recovery time (e.g., Castelle et al 2016; Keijsers et al., 2016; Ruz et al., 2017b) could introduce novel dynamics that are not explored in this work.****"*

**Page 8 Line 20-24:**

*"A warming climate might lead to further northward expansion of U. paniculata, which is currently restricted in northward extent by temperature (Seneca, 1972; Godfrey, 1977)—northern expansion of the range has already been observed (Zinnert et al., 2011; Stalter and Lamont, 1990; 2000) and is being sought in selective breeding trials (USDA, 2013).* ***Additionally, glasshouse experiments have reported that A. breviligulata is negatively impacted by competition with U. paniculata (Harris et al. 2017; Brown et al., 2017).****"*

**Page 8; line 31-35:**

"*Although beyond the scope of this effort, observational work aimed at assessing the relationships among storm frequency/magnitude, species composition of dune-building vegetation and dune development (e.g., van Puijenbroek et al., 2017a; 2017b) will be useful in addressing the future implications of model results presented here as climate change is anticipated to alter each of these factors.* "

**We also noticed an error in two of our equations: two Heaviside functions terms were inadvertently shown as 'max' functions, this has been fixed:**

**Page 3 line 26 - Page 4, line 8**

"*The intrinsic growth rate ($G_0$) is assumed to increase with the deposition rate* $max\left(\frac{dh}{dt}, 0\right)$ *and to vanish near to the shoreline* ($x < L_{veg}$ , *where $x$ is the distance to the shoreline*). **This is represented by a Heaviside function ($\Theta$) that is unity when distance to the shoreline is sufficient for plant growth $\left((x - L_{veg}) > 0\right)$, and 0 otherwise**:

$$G_0 = H_v^{-1} max\left(\frac{dh}{dt}, 0\right) \Theta(x - L_{veg}) ,$$
(2)

*The lateral vegetation propagation rate $C$ is also assumed to increase with the deposition rate and to vanish for steep slopes ($\tan\theta_c < |\nabla h|$; where $\theta_c$ is 15 degrees and is based on field observations from Moore et al., (2016).* **This is represented by a Heaviside function ($\Theta$) that is unity when the slope of the land surface is not beyond a threshold $\left((\tan\theta_c - |\nabla h|) > 0\right)$ and 0 otherwise**:

$$C = \beta\, max\left(\frac{dh}{dt}, 0\right) \Theta(\tan\theta_c - |\nabla h|) ”$$

---

## Author Comment (AC1)

**Response to Reviewer 2**

**Reviewer comments in** plain text
**Author comments are in BOLD**
**ESurfD Manuscript text is in** *italics*
**Added Text is in** ***Bold Italics***

**We thank Referee #2 for taking the time to read and review our manuscript. We address each specific comment below:**

The authors build upon their previously developed numerical ecomorphodynamic models, to demonstrate some of the factors controlling the hummockiness of a foredune, at the absence of external forcing. This is a good paper, and it would be very interesting to see the model compared to field and remote sensing observations of foredunes, so as to provide some validation to the model.

**We agree — as we stated in the last paragraph of the Discussion (which has been removed from to address comments from R1), we have ongoing monitoring work aimed at testing this model with spatially continuous vegetation and topography data, taken at regular intervals (i.e., not post-storm surveys). Although direct testing of the model is beyond the scope of the current paper, we have added a new last sentence that highlights useful next steps aimed at testing the model, and current observational research that is applicable to our modeling work:**

**Page 8; line 31-35:**

"***Although beyond the scope of this effort, observational work aimed at assessing the relationships among storm frequency/magnitude, species composition of dune-building vegetation and dune development (e.g., van Puijenbroek et al., 2017a; 2017b) will be useful in addressing the future implications of model results presented here as climate change is anticipated to alter each of these factors.*** "

Following are some more detailed comments:
The term "annealing" may be mistakenly interpreted as if a foredune is annealed and washed by waves, whereas the authors mean that the hummockiness is annealed, not the foredune. I suggest that the authors use a different term throughout the paper.

**We have replaced all uses of the term 'anneal' (referring to closing of the gap between dunes), and its variants, with 'coalesce' (referring to merging of the dunes themselves, and therefore closing of the gap).**

**We have also changed the title of the manuscript:**

***Lateral vegetation growth rates exert control on coastal foredune "hummockiness" and coalescing time***

Add a table showing all variables, abbreviations and their meaning, to make it easier for the readers to follow the equations which are developed.

**We now include a table with all variable abbreviations and names as an Appendix**

p. 2. l. 21: Continuous dune ridges may also become less continuous and hummocky with time, see: Levin, N., Tsoar, H., Herrmann, H. J., Maia, L. P., & Claudino-Sales, V. (2009). Modelling the formation of residual dune ridges behind barchan dunes in Northeast Brazil. Sedimentology, 56(6), 1623-1641.

**Thank you for pointing us to this paper. We have added text to the discussion about this:**

**Page 8 line 6-16:**

*"In addition to storms, other factors such as a high water table, low sediment supply, grain size variability, development of shell lag, and climatic conditions may also result in suppression of the coalescing of coastal foredunes (Mountney and Russell, 2006; 2009; Wolner et al., 2012; Hoonhout and de Vries, 2016;* **Ruz and Hesp, 2014; Ruz et al., 2017a***). Feedbacks between the wind, dune vegetation and sediment transport that are specific to hummocky dunes may also alter the rates of coalescing (Barrineau and Ellis, 2013; Gilles et al., 2014),* **such as the development of high wind velocity regions located adjacent to hummocky dune forms (Hesp and Smyth, 2017). Work here does not address observations of older foredune ridges that lose their continuous morphology as a result of plant succession, erosion via rain and flow in rivulets, or trampling (Levin et al 2009; 2017). Additionally the potential for lag between 'fast' cross-shore beach recovery time vs. slower cross-shore vegetation recovery time (e.g., Castelle et al 2016; Keijsers et al., 2016; Ruz et al., 2017b) could introduce novel dynamics that are not explored in this work."**

p.4 l. 12-13: Is it a reasonable assumption, that plants establish "only by lateral propagation"?

**This is an interesting issue. Though seeds are known to be a source of new *U. paniculata* plants, it is unclear to us (from our own field work and from the literature) what percentage of plants are from seed vs. lateral propagation. However lateral propagation in this model is somewhat generic, so could be inclusive of local seed dispersal and plant initiation. This is an assumption of the model. To address this question we have added to the sentence in question so that it now reads:**

**Page 4 line 24-27:**
**"In the absence of observational data that reveals the degree to which dune-building vegetation establishes via seed versus lateral propagation,** *beyond the initial 'seeding' we allow plants to establish in unvegetated cells only by lateral propagation,* **which can be thought of as encompassing establishment via both mechanisms."**

Figures 4, 5: State in the figure captions what does Hv represent.

**This has been done**

Discussion: While hummocky foredunes may indeed anneal to form continuous foredunes at their early life stages, later on, foredunes often "lose" their continuous form, as large shrubs and trees start to develop, and additional process of erosion take place. See Figure 8 in Levin, N., Jablon, P. E., Phinn, S., & Collins, K. (2017). Coastal dune activity and foredune formation on Moreton Island, Australia, 1944–2015. Aeolian Research, 25, 107-121. I also refer the authors to Castellte et al. (2017), who show that following a storm, foredune vegetation recovery time may be much longer than sand volume recovery time: Castelle, B., Bujan, S., Ferreira, S., & Dodet, G. (2017). Fore- dune morphological changes and beach recovery from the extreme 2013/2014 winter at a high-energy sandy coast. Marine Geology, 385, 41-55.

**Thank you for pointing us to these papers as well. We have added text to the discussion about the applicability to our model for only growing ridges, and the potential for novel behavior in the cross-shore direction because of the lagged timescales of beach and vegetation recovery (Keijsers et al 2016 also demonstrates this behavior in a model) :**

**Page 8 line 6-16:**

*"In addition to storms, other factors such as a high water table, low sediment supply, grain size variability, development of shell lag, and climatic conditions may also result in suppression of the coalescing of coastal foredunes (Mountney and Russell, 2006; 2009; Wolner et al., 2012; Hoonhout and de Vries, 2016;* **Ruz and Hesp, 2014; Ruz et al., 2017a***). Feedbacks between the wind, dune vegetation and sediment transport that are specific to hummocky dunes may also alter the rates of coalescing (Barrineau and Ellis, 2013; Gilles et al., 2014),* **such as the development of high wind velocity regions located adjacent to hummocky dune forms (Hesp and Smyth, 2017). Work here does not address observations of older foredune ridges that lose their continuous morphology as a result of plant succession, erosion via rain and flow in rivulets, or trampling (Levin et al 2009; 2017). Additionally the potential for lag between 'fast' cross-shore beach recovery time vs. slower cross-shore vegetation recovery time (e.g., Castelle et al 2016; Keijsers et al., 2016; Ruz et al., 2017b) could introduce novel dynamics that are not explored in this work."**